# Three months of combined high resolution rainfall and wind data collected on a wind farm 110 km South-East of Paris (France)

Auguste Gires[1], Jerry Jose[1], Ioulia Tchiguirinskaia[1], and Daniel Schertzer[1]

[1]Hydrologie Météorologie et Complexité (HM&Co), Ecole des Ponts Paris-Tech, France

**Correspondence:** Auguste Gires, auguste.gires@enpc.fr

**Abstract.** The Hydrology, Meteorology, and Complexity laboratory of École des Ponts ParisTech (hmco.enpc.fr) has made a data set of high resolution atmospheric measurements available, which is of interest for the atmospheric science community. It comes from a campaign carried out, in the framework of the RW-Turb project (supported by the French National Research Agency – ANR-19-CE05-0022), on a meteorological mast installed at a wind farm located approx. 110 km South-East of Paris in France. Three months covering the spring period from 1/3/2021 to 1/6/2021 are made available. Six devices are used: two 3D sonic anemometers (manufactured by Thies), two mini meteorological stations (manufactured by Thies), and two disdrometers (Parsivel[2], manufactured by OTT). They are installed at two heights (approx. 45 m and 80 m), which enables to monitor potential effects of altitude. The temporal resolution is of 100 Hz for the 3D sonic anemometers, 1 Hz for the meteorological stations and 30 s for the disdrometers. A multifractal analysis is implemented to assess the effective resolution of the devices and it suggested that anemometers and stations are able to measure expected variability only down to 1 s and 16 s respectively. Link to the data set: https://doi.org/10.5281/zenodo.5801900 (Gires et al., 2021)

## 1  Introduction

A limited number of studies have investigated the effect of rainfall on wind turbines and they tend to indicate that it is rather significant. For example, Corrigan and Demiglio (1985) reported a power production decrease of 20 to 30% from an experiment conducted in Ohio (USA) on a 38 m diameter two blade wind turbine, with a greater worsening with greater rain rates. Walker and Wade (1986) slightly disputed those results for light rain (i.e. rain rate smaller than 8 mm.h$^{-1}$). In fact, on the contrary, they found an increase of few % for light rain and attributed it to modifications of blade roughness or wind measurement issues. The significant decrease was later confirmed experimentally (Al et al., 1986) and with multiphase (volatile for air and liquid for rain) computational fluid dynamics (Cai et al., 2013; Cohan and Arastoopour, 2016).

Understanding the effect of rainfall on wind power production is hence highly relevant. The Rainfall Wind Turbine or Turbulence project (RW-Turb), which is supported by the French National Research Agency (ANR in French), actually aims at contributing to the topic. The data presented in the paper was collected in the framework of this project. In order to properly address the topic, two distinct aspects should be studied properly; first the rainfall effect on the energy resources and second the rainfall effect on the conversion process of wind power to electric power by the wind turbine. Towards this, rainfall should not only be understood, as commonly done, as a simple rain rate expressed in mm.h$^{-1}$, but also by considering its full complexity

through the spatial and temporal variability of the drop size distribution (DSD). The data set shared is especially tailored for this point. Indeed, both the wind energy and torque available to wind turbines are basically proportional to powers of the instantaneous wind speed. As wind is neither constant nor uniform, taking into account its small scale spatio-temporal fluctuations is crucial to properly quantify the integrals of these quantities especially given that the wind turbines are located in the atmospheric boundary layer, which is an area of increased complexity due to the interactions with the ground. Improving turbulence understanding has been listed among the scientific challenges of this field in a recent joint paper by leading academics of the field (van Kuik et al., 2016), for the European Academy of Wind Energy, EAWE). The intrinsic intermittent nature of wind, i.e. the fact that its activity becomes located on smaller and smaller support as observation scale decreases, makes it complex to analyse, notably requiring appropriate theoretical framework and high resolution measuring devices. An illustration of this can be found in Fitton et al. (2011, 2014) in which authors studied 3D wind data collected from two different locations (Corsica and Germany) in a multifractal framework enabling to highlight the need to investigate turbulence in a 3D framework so that the anisotropy between the horizontal and vertical wind components is accounted for. They also pointed out that such scale invariant framework is needed to explain the power law fall-off for the probability distribution of wind fluctuations and to account for the observed sporadic bursts, which are not treated in a standard Gaussian framework that strongly underestimates the extremes. Rainfall was not considered in such analysis, up to now.

Hence, given the numerous potential applications of combined high resolution rainfall and wind data, notably in the framework of wind energy production, the Hydrology, Meteorology, and Complexity laboratory of École des Ponts ParisTech (HM&Co-ENPC) considers that it is relevant to make the data available to the scientific community, from a 3-month (1/3/2021 - 1/6/2021) measurement campaign carried out on a meteorological mast operated by Boralex, a wind power producer. The campaign involves 6 devices: a 3D sonic anemometer, a disdrometer (which gives access to size and velocity of drops falling through its sampling area) and a mini meteorological station located at roughly 78 m; the same setting is repeated at roughly 45 m. The devices' functioning as well as the measurement campaign is presented in section 2. The corresponding database and available tools are presented in section 3.

Before proceeding further, the purpose of the paper should be clarified to avoid any misunderstanding. It is a data paper that aims at presenting, in details, a data set made available to the community. It does not aim at fully exploiting the data set for scientific studies; this will be done in further dedicated papers by the authors or community members using it. The data set was collected in a framework designed for application in wind energy, but potential applications of such high resolution rainfall and wind measurement campaign are much wider. For example, understanding rainfall processes remain a major challenge in the field of hydrology. The lack of precise space-time distributed measurement is one of the greatest sources of uncertainty in hydrological modelling. Improving understanding of rainfall requires an in-depth understanding of its relationship to wind turbulence, across scales. This could notably lead to the development of 3D+1 model for drops' location, which could be used to overcome the simplistic assumption of homogeneous distribution within a radar gate (see Gires et al., 2016, and references therein for an initial discussion on the topic) or to improve wind drift correction scheme for radar algorithms. Such developments will improve precipitation estimation with the help of radars.

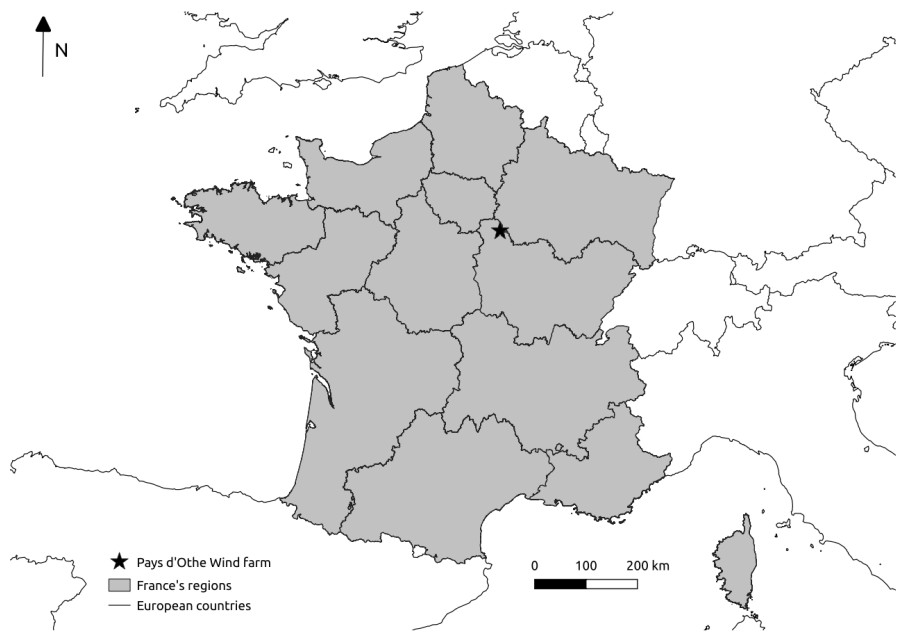

**Figure 1.** Location of the Pays d'Othe wind farm in France, where the data presented in this paper is collected from

## 2  Data and methods

### 2.1  Location of the meteorological mast

The devices involved in this measurement campaign are installed on a meteorological mast located on a wind farm at Pays d'Othe, France. This wind farm is made of 9 wind turbines and is jointly operated by Boralex (https://www.boralex.com/our-projects-and-sites/) and JP Énergie Environnement (https://pays-othe-89.parc-eolien-jpee.fr/). It is located at roughly 110 km South-East of Paris, on the territory of the cities of Vaudeurs, Coulours, Les Sièges (see Fig. 1). Figure 2 displays a zoomed map of the surroundings with an OpenStreetMap background. The meteorological mast is the star in the middle. The nine wind turbines of the Pays d'Othe wind farm are aligned South-East of it and within a 4 km radius (black vertical crosses). They are visible on the left and right pictures of Fig. 3. The five turbines of the Molinons wind farm in the North are also visible within the 5 km radius (grey vertical crosses). It should also be noted that a small grove is located just South of the mast at roughly 160 m. A larger one is on the East at roughly 100 m. These groves are also visible on the middle picture of Fig. 3.

Figure 4 displays the elevation around the meteorological mast. The data used is the IGN DBALTI 75M. It has a horizontal resolution of 75 m and is provided by the National Institute of Geographic and Forest Information (IGN). The elevation of the pixel where the mast is located is 230 m. Nearby the mast (i.e. within the 1 km radius), there is a small slope in the North-South direction, which is visible on the right picture of Fig. 3.

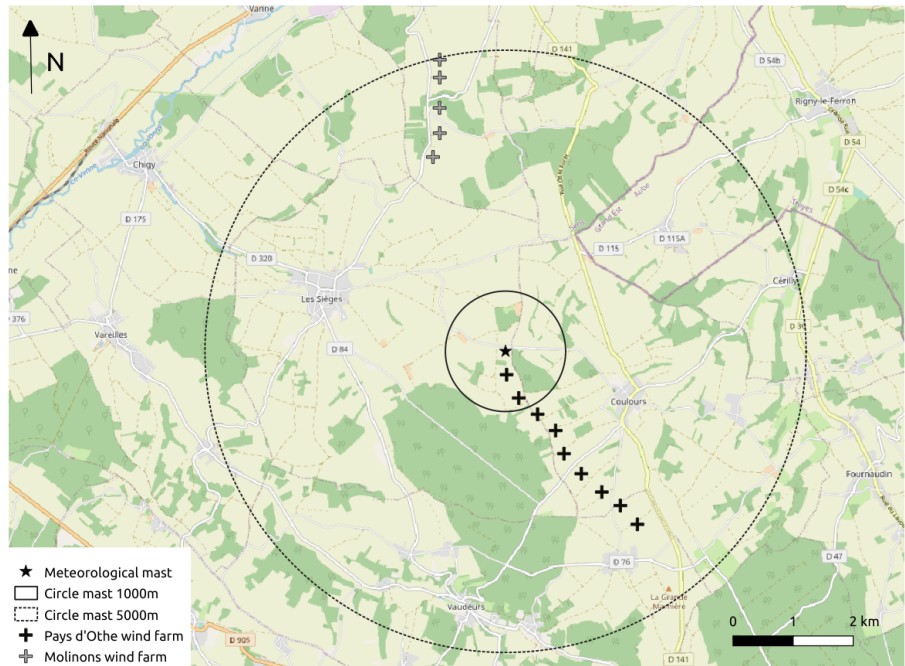

**Figure 2.** Map of the surroundings of the meteorological mast used. The background of the map is taken from OpenStreetMap (https://www.openstreetmap.org/, © OpenStreetMap contributors 2021. Distributed under the Open Data Commons Open Database License (ODbL) v1.0.). Forest are in green, farms are in light green, residential areas are in grey, roads are in white (small ones) or orange (highway).

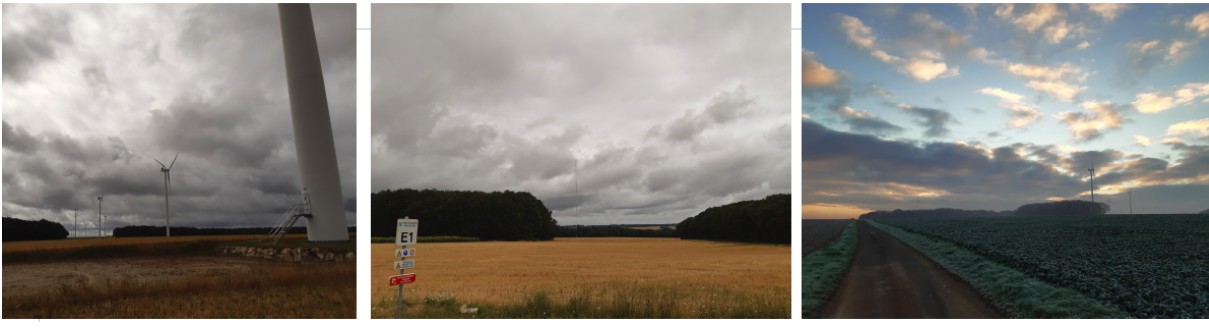

**Figure 3.** Pictures of the Pays d'Othe wind farm. Left: picture of the wind farm taken from the wind turbine closest to the mast. Middle: picture of the meteorological mast taken from the wind turbine closest to it. Right: picture of the mast (left of the picture) and the wind farm (right of the picture) taken from the road just North of the mast. Pictures were taken by A. Gires.

Figure 5 exhibits a picture of a meteorological mast along with the six devices installed on it. More precisely, at approximately 78 m, a 3D sonic anemometer, a mini meteorological station and a disdrometer are installed. The same setting is repeated at roughly 45 m. The precise elevation and offset from the mast of the six devices are indicated on near the corre-

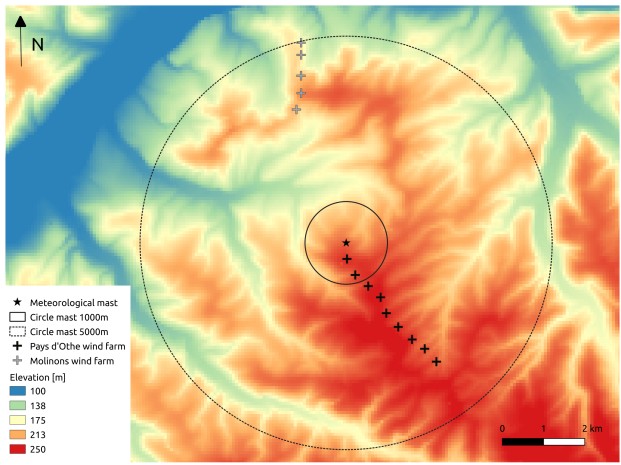

**Figure 4.** Elevation map around the Pay d'Othe wind farm. For the elevation in m, the IGN DBALTI 75M product is used.

sponding zoomed pictures on Fig. 5. The two raspberry pi computers, which are collecting data along with the 4G box enabling to access data remotely, are located in one of the boxes at roughly 10 m.

## 2.2 3D sonic anemometers and associated outputs

Two 3D sonic anemometers manufactured by ThiesCLIMA (ThiesCLIMA, 2013a) are used in this measurement campaign. A 3D sonic anemometer is made of three pairs of transducers. Let us denote $L$ as the distance between two transducers and $u_L$ as the wind velocity along the corresponding axis. The transducers can be either transmitters or receivers of a sound pulse, and they constantly swap roles. It means that the device actually measures the travel time of a pulse of sound between the two transducers in one way or the other. If these times are denoted by $t_1$ and $t_2$, we have $t_1 = L/(c + u_L)$ and $t_2 = L/(c - u_L)$, with $c$ being the local speed of sound in the air; this yields:

$$u_L = \frac{L}{2}\left(\frac{1}{t_1} - \frac{1}{t_2}\right) \tag{1}$$

which does not depend on $c$. The wind velocity is assessed along the axis between each three pairs, enabling to reconstruct 3D wind.

It is also possible to estimate $c$ from:

$$c = \frac{L}{2}\left(\frac{1}{t_1} + \frac{1}{t_2}\right) \tag{2}$$

Since $c$ mainly depends on the local temperature $T$, the latter can be derived using standard relationships assuming a dry air. This yield a virtual sonic temperature. Additional correction can be implemented to derive a corrected temperature accounting for relative humidity and pressure (see ThiesCLIMA, 2013a for more details). Hence, the 3D anemometers provide 3D wind measurement along with an estimate of temperature. The sampling rate used in this campaign is of 100 Hz for these devices.

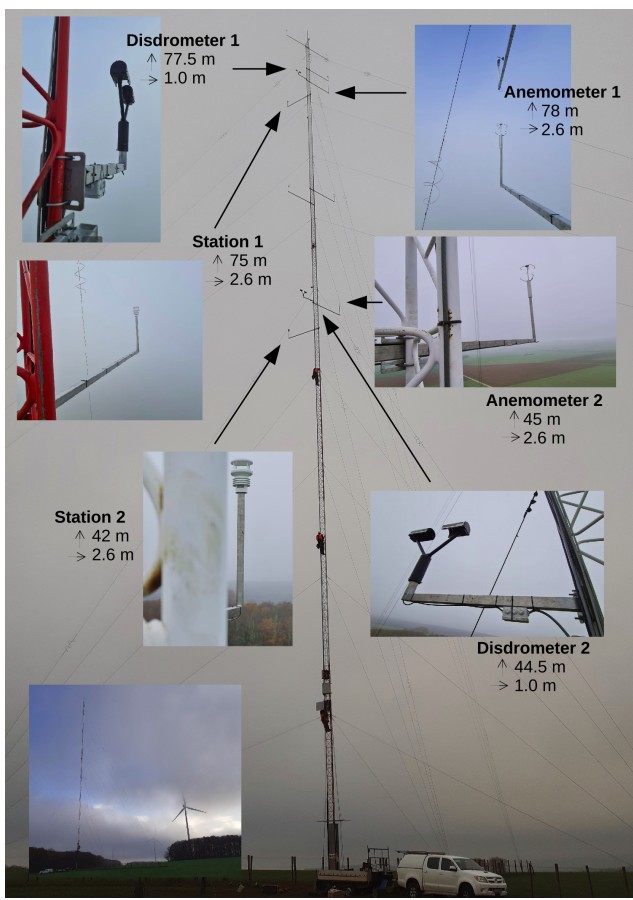

**Figure 5.** Summary of the measurement device's location on the meteorological mast

## 2.3 Meteorological station and associated outputs

Two mini meteorological stations manufactured by ThiesCLIMA (ThiesCLIMA, 2013b) are used in this measurement campaign. They give access to the most relevant meteorological parameters: wind velocity and direction, air temperature, relative humidity, precipitation and brightness. The sampling rate used in this campaign is of 1 Hz for these devices.

The wind information is obtained thanks to a 2D sonic anemometer made of two pairs of transducers positioned perpendicularly in relation to each other. See section 2.2 for more details on the functioning of such device. Built-in sensors are dedicated to measurement of air temperature and relative humidity. The measurement of pressure relies on a micro-electro-mechanical system. The latter three are protected within a shelter. Precipitation intensity is estimated with the help of a mini Doppler radar. The signal reflected back by the hydrometeor is analysed and a rain rate is derived relying on strong assumptions of the DSD

shape and the relation between size and velocity of drops. Brightness is measured with the help of four photo sensors, whose spectral sensitivity curve is tuned to the human eye's sensitivity. In addition, there is a GPS sensor.

## 2.4 Disdrometer and associated outputs

Two OTT Parsivel[2] disdrometers (OTT, 2014) are used in this measurement campaign. OTT is the name of the manufacturer. Such device gives access to the size and velocity of the drops passing through its sampling area. The data regarding disdrometer is actually similar to the one already discussed in Gires et al. (2018). Hence, the interested reader is referred to this paper and references therein for a detailed presentation of the devices and associated output. It will only be reminded here that the main output of the disdrometer is actually a matrix with the number of drops recorded ($n_{i,j}$) during the time step $\Delta t$ (30 s here) according to classes of equivolumic diameter (index $i$ and defined by a centre $D_i$ and a width $\Delta D_i$ expressed in mm) and fall velocity (index $j$ and defined by a centre $v_j$ and a width $\Delta v_j$ expressed in m.s$^{-1}$). From this matrix, it is possible to derive any rainfall related quantity and notably:

- The rain rate $R$ (in mm.h$^{-1}$)

- The drop size distribution (DSD) $N(D)$ (in m$^{-3}$.mm$^{-1}$) $N(D)dD$ is the number of drops per unit volume (in m$^{-3}$) with an equivolumic diameter between $D$ and $D + dD$ (in mm). Given the binned nature of the disdrometer data, it is a discrete DSD, denoted $N(D_i)$ that is actually computed. Many quantities relevant to researchers and practitioners can actually simply be expressed as moments of the DSD.

The table for classes of diameter and velocity, the formulas for computing $R$ and $N(D_i)$ as well as associated moments, and the filters imposed can be found in Gires et al. (2018) and are therefore not repeated here. Disdrometers have been widely used to measure rainfall, and examples of measurements comparison between themselves and with rain gauges can be found in Miriovsky et al. (2004), Krajewski et al. (2006), Frasson et al. (2011) or Thurai et al. (2005), which are some examples among others.

## 2.5 Measurement period

The data presented in this paper was collected between 1 March 2020 and 31 May 2020. Over this three months period, there is a limited number of missing time steps with only 3840, 4237, 3691, 3734, 3658, 3658 missing minutes for respectively Anemometer #1, Anemometer #2, Station #1, Station #2, Disdrometer #1 and Disdrometer #2. In the worst case, it corresponds to approximately 3 % of the time. They mainly correspond to periods of power cut on the mast which turns off the devices and the computer retrieving the data. They are mainly short (few minutes) power cuts except on 8-10 March during which the power remained cut for more that 2 days. Few more minutes are missing with the anemometers. This is likely because of the sporadic loss of information by the retrieving computer due to high sampling frequency.

Figure 6 displays the rain rate and cumulative rainfall depth vs. time during the 3 month period (first row). The two disdrometers give very close estimates (159 mm for #1 and 157 mm for #2) with a difference smaller than 1.6 %. The two meteorological stations have a stronger difference between (119 mm vs. 130 mm, hence roughly 9% difference) them and yield significantly smaller total rainfall depth. The difference, on average, is 24 % between the disdrometers and the stations. Such difference is likely to be due to the fact that both devices rely on completely different measurement techniques. It should definitely be

**Figure 6.** Rain rate (upper left) and cumulative rainfall depth (upper right) vs. time over the three months of measurement campaign, retrieved from the two disdrometers and the two meteorological stations. 1 min average wind (lower left) and wind rose (lower right) computed from the 3D sonic anemometer #1 which is located at the top of the mast.

explored further in future investigations. The 1-min average total horizontal wind from the 4 measuring devices is displayed
in the lower left panel of Fig. 6. In the area, daily average wind is higher in winter with values up to 7.5 m.s$^{-1}$ and lower in
summer with minimum values of 4.5 m.s$^{-1}$. These values were obtained using 30 years of 50 m height wind from the MERRA
(Modern-Era Retrospective Analysis for Research and Applications) data, which is a NASA reanalysis (Bosilovich et al., 2016;
Gelaro et al., 2017). The available period has a average wind of 6 m.s$^{-1}$, which is consistent with usual values, although such
average fully neglects the variability which is what this data set enables to study. Finally, the wind rose for the entire period is
on right panel of Fig. 6. It can be seen that the wind is mainly oriented along a South-West / North-East axis.

## 3 Database

In this section, a detailed description of the database content and of some available basic scripts is provided. The overall
organization is first described before providing some details on the 'Calendar_RW_Turb_wind_farm' folder, and the format of
the files for each devices.

## 3.1 Overall organization

The structure of the database is actually inherited from the data management flow which is basically similar for all the devices:

- Raw data is initially collected in the form of .txt files corresponding directly to the outputs of the device. It is stored in files corresponding to 1 min of data for stations and anemometers, and 30 s for disdrometers.

- All these files are then zipped for individual days and stored in a corresponding folder in 'Raw_data_zip/'

- These raw files are then used to generate daily (30 s or 1 Hz) or hourly (100 Hz for anemometer) files in an "easy to read" format. .csv files, then zipped to limit their size) are used for stations and anemometers. .npy files are used for parsivel. They are stored in the corresponding folders of the database whose name is quite explicit.

- It is these files and not the raw ones that are used by the python scripts to extract the data according to the user's needs.

Hence, the following structure was adopted for the database :

Data_base_rw_turb/

    Raw_data_zip/

        Anemometer_1/

        Anemometer_2/

        Station_1/

        Station_2/

        Pars_RW_turb_1/

        Pars_RW_turb_2/

        Each folder contains the files for its devices.

        The name is Raw_DeviceName_YYYYMMDD.zip (ex: Raw_Anemometer_20210318.zip).

    Daily_data_python_disdrometer/

        Pars_RW_turb_1/

        Pars_RW_turb_2/

        Each folder contains the files for its disdrometers.

        The name is DisdroName_raw_data_YYYYMMDD.npy (ex : Pars_RW_turb_1_raw_data_20210318.npy).

    Daily_data_1Hz/

        Anemometer_1/

        Anemometer_2/

        Station_1/

        Station_2/

        Each folder contains the files for its device.

        The name is Daily_data_1Hz_DeviceName_YYYYMMDD.csv (ex: Daily_data_1Hz_Station_2_20210318.csv), which is zipped after.

Hourly_data_100Hz/

    Anemometer_1/

    Anemometer_2/

    Each folder contains the files for its device.

    The name is Hourly_data_100Hz_DeviceName_YYYYMMDD_HH.csv

    (ex: Hourly_data_100Hz_Anemometer_2_20210318_07.csv), which is zipped after.

Calendar_RW_Turb_wind_farm

    Data_30_sec/ (one file per day ex: R_30_sec_RW_Turb_wind_farm_2021_03_18_00_00_00__2021_03_19_00_00_00.csv)

    Quicklooks/ (one file per day ex: Quicklook_RW_Turb_wind_farm_2021_03_18_00_00_00__2021_03_19_00_00_00.png)

    Calendar_RW_Turb_wind_farm.html

Python_scripts/

    The python scripts (and associated files) to generate and use this database are located in this folder.

Read_me.txt

    It contains a short description of the RW-Turb database.

## 3.2 Calendars

The purpose of this folder is to provide the user with a rapid access to a visual overview of the available data and enable him/her to easily identify relevant periods / days according notably to the rainfall conditions. This is done through an .html file ('Calendar_RW_Turb_wind_farm.html') which contains links to daily quicklooks and provides a rapid overview of the measurement campaign. Fig. 7 displays a snapshot of it. Since mentioned links are relative ones, the file should be located as indicated in the above structure for a proper functioning. Fig. 8 shows an example of daily quicklook. It provides a summary of the recorded weather conditions with (explained from top to bottom and left to right in a given row):

- Panel (a): Rain rate vs. time.

- Panel (b): Cumulative rainfall depth vs. time.

- Panel (c): Indication of the missing data if any. Each line corresponds to a device. One minute time steps are used.

- Panel (d): DSD $N(D)$ vs. time.

- Panel (e): Total horizontal wind ($\sqrt{u_x^2 + u_z^2}$ vs. time for the various available devices. One minute time steps are used.

- Panel (f): Temperature (°C) vs. time with the various available data. One minutes time steps are used.

- Panel (g): A representation of the number of drops according the velocity and size classes for the whole event. A solid black line corresponding to a standard relation (Lhermitte, 1988) between the terminal fall velocity of drops and their equivolumic diameters was added.

- Panel (h): A wind rose using the horizontal wind measurements ($u_x$ and $u_y$) of the two 3D sonic anemometers.

- Panel (i): Pressure (hPa) vs. time from the two meteorological stations. One minutes time steps are used.

- Panel (j): $N(D)D^3$ as a function of $D$ (lower left). $N(D)D^3$ is plotted and not simply $N(D)$ because it is basically proportional to the volume of rain obtained according to the drop diameter. Hence it provides the reader with a better immediate insight of the influence of the various drops size on the observed rainfall event).

- Panel (k): Vertical wind ($u_z$) vs. time from the two 3D sonic anemometers. One minutes time steps are used.

- Panel (l): Relative humidity (%) vs. time from the two meteorological stations. One minutes time steps are used.

All the daily quicklooks are stored in the folder Quicklooks/ and can be accessed directly there. To ease access, the file name is simply Quicklook_RW_Turb_wind_farm_ followed by the date of start and end of the corresponding day in string format. For example, the one displayed in Fig. 8 is called 'Quicklook_RW_Turb_wind_farm_2021_05_06_00_00_00__2021_05_07_00_00_00.png'. Local times are used.

Finally, the folder Data_30_sec/ contains daily rain rate files with the rain rate (in mm.h$^{-1}$) stored for each 30 s time step of
the day in .csv format. They are named in a similar way as the quicklooks
(example : 'R_30_sec_RW_Turb_wind_farm_2021_03_18_00_00_00__2021_03_19_00_00_00.csv'). The format is straightforward: (i) One line per 30 $s$ time step starting on YYYY-MM-DD 00:00:00 (UTC time); (ii) In each line, values for the two disdrometers are separated with semicolumn (the order is Pars#1;Pars#2); (iii) Missing data are noted as "nan".

## 3.3   3D anemometer data

The raw data is made of one .txt files per minute containing directly the output telegram of the device. They are called Raw_DeviceName_YYYYMMDD_HHMM.txt Such file is made of 6000 lines, corresponding to 1 min at a sampling rate of 100 Hz, having this format:

b'02+000.03;-000.02;-000.01;+22.9;0E;47°03' where the 3D wind is given, followed by the virtual sonic temperature and two status codes of the device. These files are then zipped per day and stored in the corresponding folder Raw_data_zip/.

Finally, some .csv files containing all the relevant data are generated for each device. Hourly files are used for the data at 100 Hz and daily files for the data at 1 Hz (obtained simply be averaging 100 successive time steps). They are then stored in the corresponding folder of the database (see section 3.1) in a zipped format to reduce their size.

The format is straightforward with one line per each time step (0.01 s or 1s) and the four quantities of interest separated with semicolon, i.e $u_x$ (m.s$^{-1}$); $u_y$ (m.s$^{-1}$); $u_z$ (m.s$^{-1}$); T (°C). The 'x' axis is oriented toward the East of the device, the 'y'
toward the North and the 'z' upward. Missing data are noted as 'nan'.

**RW-Turb campaign on Pays d'Othe wind farm**

**Quicklooks**

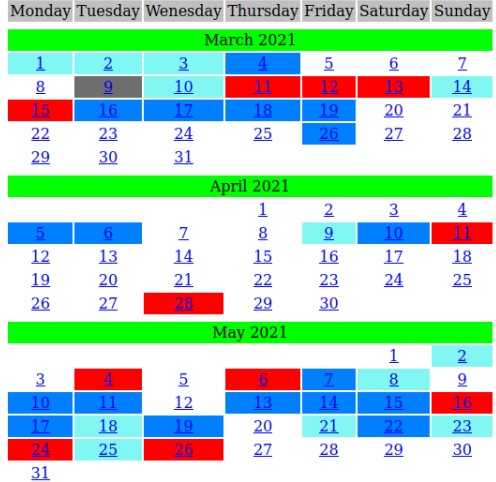

**Color code**

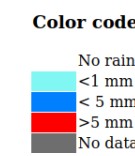

No rain
<1 mm
< 5 mm
>5 mm
No data

**Figure 7.** Snapshot of the calendar summarizing the March-May 2020 measurement campaign on the meteorological mast of the Pays d'Othe wind farm.

## 3.4 Meteorological station data

The raw data is made of one .txt files per minute containing directly the output telegram of the device. They are called Raw_DeviceName_YYYYMMDD_HHMM.txt Such file is made of 60 lines, corresponding to 1 min at a sampling rate of 1 Hz, having this format:

b'00.01;000.0;+24.6;21062;21043;21060;21043;99;0;+23.0;+24.5;030.0;030.8;1025.8;000382;000809;000329;000282;000809;
000;0000.569;1;+25.1;35.9;7297003;+48.842293;+002.588063;0148;038.7;169.2;13.03.20;11:25:47*13°'

where all the data is reported, separated by semicolon (see next paragraph for the order). These files are then zipped per day and stored in the corresponding folder Raw_data_zip/, to reduce their size.

Finally, some daily .csv files containing all the data are generated for each device. They are then stored in the corresponding

folder of the database (see section 3.1). The format is : (i) One line per 1 $s$ time step starting on YYYY-MM-DD 00:00:00 (UTC time); (ii) In each line, values of measurement are provided separated by semicolon; (iii) Missing data are noted as "nan". In each line, the order of the data is :

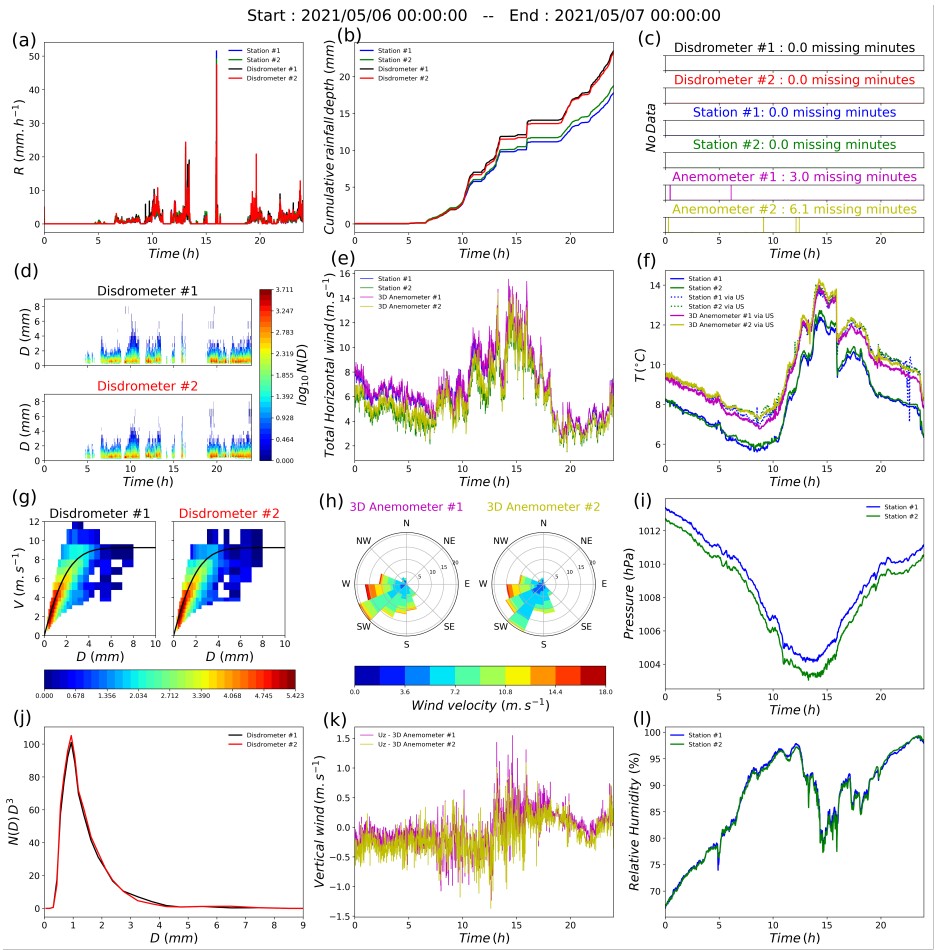

**Figure 8.** Quicklook of the meteorological data available on 6 May 2021. Precise description of each panel in given in the text.

- Wind speed [m/s]

- Wind direction [°], (starting clockwise from the North of the device)

- Virtual temperature [°C]

- Propagation time converter 3 towards converter 1 (south to north)

- Propagation time converter 4 towards converter 2 (west to east)

- Propagation time converter 1 towards converter 3 (north to south)

- Propagation time converter 2 towards converter 4 (east to west)

- Measured value buffer content level 0...99

- Heating requirement

- Calculated air temperature [°C]

- Temperature uncompensated [°C]

- Relative humidity uncompensated [%]

– Calculated relative humidity [%]

- Air pressure [hPa]

- Brightness north [lux]

- Brightness east [lux]

- Brightness south [lux]

– Brightness west [lux]

- Brightness max. value / vectorial sum [lux] (s. Command BO)

- Direction of brightness [°]

- Precipitation intensity [mm/h]

- Precipitation event [0/1]

– Temperature in housing [°C]

- Supply voltage [V]

- Internal counter [ms]

- Latitude [°]

- Longitude [°]

– Height of sensor referred to sea level [m]

- Position of the sun, elevation [°] (-90°...+90°= zenith)

- Position of the sun, azimuth [°] (0° = north ; 180°= south)

## 3.5 Disdrometers data

As for the other devices, the raw data is made of one .txt file per 30 s time step containing all the collected data. It should be
stressed that it corresponds to the raw data and is made available for expert users only. Most users will not need to access them
and will be satisfied with the provided python scripts. The precise format for these raw fields can be found in the Python scripts
with the heading of corresponding functions.

Finally, daily file containing a list of the data collected by the devices for each time steps are generated and stored on the
corresponding folder of 'Daily_data_python_disdrometer/'. They are stored in .npy format and are readable with the help of
Python 3. The Python scripts made available actually use these files. More precisely, each element of the list corresponds to a
time step of 30 s. Each element of the list is again a list containing these elements (in the same order):

- 0 : Sensor ID

- 1 : Precipitation rate (mm/h), computed by the device

- 2 : Temperature in the sensor (°C), which is a rough estimate used for control than as a meteorological measurement

- 3 : OTT standard size/velocity map, i.e. a 32 x 32 matrix containing the number of drops recorded according to classes
  of size and velocity. Rows correspond to classes of size while columns correspond to classes of velocity.

## 3.6 Python_scripts/

This section presents some Python scripts that are designed to enable the user carry out some initial analysis as well as basic data
treatment with the database. The functions, which are basic tool boxes, can be found in the script 'Tools_overall_management_RW_Turb_da
The main functions are (only a short description is given here; more details, including precise description of the inputs and
outputs of the functions, are provided as comments inside each script):

- Quicklook_RW_Turb_wind_farm : it generates a quicklook image and the corresponding 30 s and 5 min rain rate time
  series for a given rainfall event during the measurement campaign on the wind farm.

- extracting_one_event_Parsivel: it reads daily.npy files and generates a list containing all the data that can be analyzed.

- extracting_one_event_anemometer_100Hz: it reads daily files at 100 Hz and generates a matrix with the data for a given
  anemometer and event.

- extracting_one_event_anemometer_1Hz: it reads daily files at 1 Hz and generates a matrix with the data for a given
  anemometer and event.

- extracting_one_event_station_1Hz: it reads daily files at 1 Hz and generates a matrix with the data for a given station
  and event.

Examples for use of the various functions can be found in the scripts 'Script_overall_management_parsivel_v1.py', 'Script_overall_management_station_v1.py', 'Script_overall_management_anemometer_v1.py', and 'Script_overall_management_RW_Turb_campaign_v1.py'. On a standard laptop, it typically takes few seconds to extract and display all the data for one day.

## 4  Effective resolution of the data

While studying the small scale space-time fluctuations, it is advantageous to use data at the finest available resolution. However, it is possible that the actual sampling resolution may be different due to quality problems in the series, leaving spurious estimates at finer scales. To understand this, 100 Hz data from anemometers and 1 Hz data from meteorological station were analysed using spectral analysis and the framework of Universal Multifractals (UM).

Spectral analysis is a commonly used technique in turbulence to estimate scaling behaviour using second order statistics. In the case of scaling behaviour, power spectrum $E(k)$ and frequency are power law related :

$$E(k) \approx k^{-\beta} \tag{3}$$

where $k$ is the corresponding frequency and $\beta$ is the spectral exponent (slope in log-log plot).

In UM framework, all moment orders are used and not only the moment order 2 as in spectral analysis. This enables to capture information over a larger spectrum - from higher order moments which emphasise on extremes to small order moments focusing on smaller values. In this framework we have:

$$< \epsilon_\lambda{}^q > \approx \lambda^{K(q)} \tag{4}$$

where $\epsilon_\lambda$ is the studied conservative field at resolution $\lambda$ (i.e. the ratio of largest scale to observation scale) and $K(q)$ the moment scaling function at moment order $q$ (Schertzer and Lovejoy (1987), or Schertzer and Tchiguirinskaia (2020) for a recent review). $K(q)$ fully characterizes the variability across scales of the studied field. In UM framework, $K(q)$ is fully determined with the help of only two 'Universal Multifractal' parameters - mean intermittency co-dimension $C_1$ and multifractality index $\alpha$. $C_1$ measures mean intermittency in the field; when $C_1 = 0$ the field is homogeneous with little variability. $\alpha$ quantifies how much this intermittency changes when moving away from the average behaviour. $0 \leq \alpha \leq 2$; higher the value of $\alpha$, higher the variability, with $\alpha = 0$ being a monofractal field where intermittency of extreme is same as that of mean. A multifractal analysis of collected data is performed to check for the effective resolution of the data, i.e. to assess if measurements are affected or not by instrumental artifacts at small scales. Given the stated purpose, only small scales (i.e. from 16 s down to 0.01 s) are studied here. Analysis and interpretation of larger scales regimes will be carried out in further scientific papers.

Spectral analysis which consists of plotting Eq. 3 in log-log and trace moment (TM) analysis which consists of plotting Eq. 4 in log-log for various moments $q$ enable to confirm scaling behaviour of studied fields. It is the case if straight lines are retrieved, potentially with several scaling regimes. The retrieved slopes give $\beta$ for the spectral analysis and $K(q)$ in the TM analysis. In fig. 9a, trace moment (TM) analysis, and spectral analysis for 100 Hz anemometer data is shown (ensemble

analysis of 1 month long data - 01/03/2021 to 01/04/2021 - with a sample length of 40 minutes). Other periods have been tested and yielded similar results with regards to the effective resolution issue discussed in the framework of this data paper. A spectral spike is observed at frequency $0.0304 \text{ s}^{-1}$ and spurious fluctuations are visible for small scales. The spike is due to the fact that at 100 Hz, same data is basically repeated over three successive time steps. Estimates of UM parameters, obtained with the help of Double Trace Moment analysis (a more robust form of TM tailored for UM fields) yield for the small scale regime (1 Hz - 100 Hz) values of $C_1$ too low ($2.80 \times 10^{-5}$) to consider any variation in the field. As the field is too smooth here (high value of $\beta$: 2.13 and 1.57), as suggested by Lavallee et al. (1993), fluctuations were analysed by differentiating the field. This enables to study a approximation of the underlying conservative fields (hence the decrease in estimates of $\beta$ and $H$). In fluctuations of same 100 Hz data, nearly 70% of the values are equal to zero, which results in strong bias for estimates with an artificial decrease of $\alpha$ ($= 0.31$ here) and an increase in $C_1$ ($= 0.21$ here), which is consistent with bias associated with numerous zeros (Gires et al. (2012). This further suggests the possibility of having instrumental noise in resolutions finer than 1 Hz. It is unclear where exactly the scaling break is (close to 1 Hz or 10 Hz) to consider instrumental noise, but for being on the safer side, we decided to take 1 Hz as the limiting value. Analysis of fluctuations of 1 Hz data (ensemble analysis of 1 month long data - 01/03/2021 to 01/04/2021 - with a sample length of 16 hours) is shown in fig. 9b. For the small scale regime (1 s - 16 s), we find $\alpha = 1.49$ and $C_1 = 0.09$ which is more consistent with estimates commonly retrieved for atmospheric fields.

Similar results (extremely small values of $C_1$ or $\beta$ suggesting instrumental noise) are observed for other 1 Hz data available at meteorological stations - Temperature ($T$), Pressure ($P$), Humidity ($RH$) and air density ($\rho$, a function of $T$, $P$ & $RH$) with 16 s being close to the actual effective sampling resolution. Fig. 9c shows the TM analysis for $T$; on the basis of spectra, the second scaling regime (16 s to 1 Hz) seems to suggest presence of instrumental artifacts (ensemble analysis of 1 month long data - 01/03/2021 to 01/04/2021 - with a sample length of 16 hours). For the 1 s - 16 s regime, we find $\alpha = 1.99$ and $C_1 = 1.61 \times 10^{-6}$; the low $C_1$ supports spectral observation. In 1 Hz station data, values of many data points were actually very close to each other resulting again in the presence of a lot of zeroes in fluctuations of the series (about 75% for $T$ fluctuations). This in turn gave biased estimates of both $\alpha$ and $C_1$. Averaging data over time reduced this effect and by considering fluctuations of data at 15 s, realistic values of $\alpha$ and $C_1$ were retrieved (Fig. 9d; $\alpha = 1.12$ and $C_1 = 0.14$ for 15 s - 4 min scaling regime).

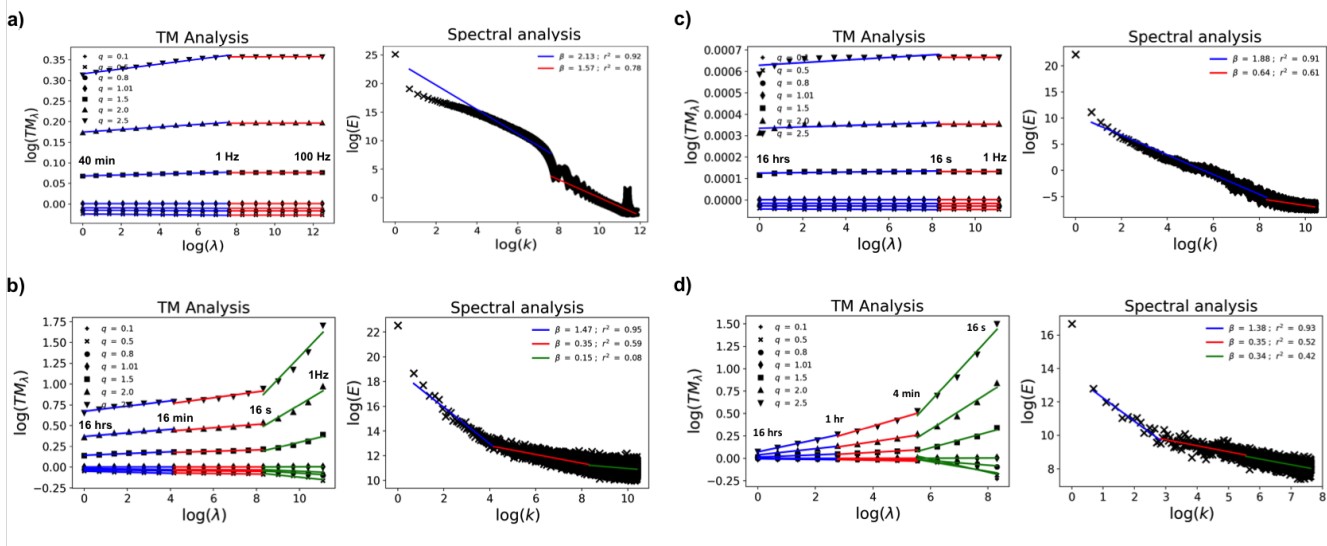

**Figure 9.** TM analysis (Eq. 4 in log-log plot), and spectral analysis (Eq. 3 in log-log plot) of 1 month long data (27/01/2021 to 27/02/2021) for a) anemometer data at 100 Hz (sample length of 40 min), b) fluctuations of anemometer data at 1 Hz (sample length of 16 hours), c) Temperature ($T$) at 1 Hz (sample length of 16 hours), d) fluctuations of Temperature ($T$) at 15 s (sample length of 16 hours)

For analysing the fields' variability, its worthwhile to note that the actual sampling resolution - resolution from which fields can be studied to obtain consistent UM parameters - is not necessarily the lowest resolution of instrumental data availability. Indeed, it could be affected by instrumental artifacts (white noise, repeated values). Here, it is more realistic to study anemome-370 ter as well as station data at a coarser resolution (1 Hz and 16 s respectively) where it is exhibiting clear scaling variability than at the finest available resolution of data recording (100 Hz and 1 Hz). Multifractal framework is a powerful tool to study this issue and assess the quality of the data.

## 5 Data availability

The data from a three month measurement campaign with devices installed on a meteorological mast of Boralex located at the 375 wind farm of Pays d'Othe is presented in this paper. Raw data along with python formatted data with corresponding scripts are described. The Hydrology, Meteorology and Complexity laboratory of Ecole des Ponts ParisTech (HMCo-ENPC) has made this data set available at https://doi.org/10.5281/zenodo.5801900. The following citation should be used for every use of the data :

– For this paper : INCLUDE CITATION of this paper

– For the database : Gires, Auguste, Jose, Jerry, Tchiguirinskaia, Ioulia, & Schertzer, Daniel. (2021). Data for : "Three months of combined high resolution rainfall and wind data collected on a wind farm" [Data set].

Zenodo. https://doi.org/10.5281/zenodo.5801900 (Gires et al., 2021)

This data set is available for download free of charge. License terms apply. The campaign is actually still ongoing. Regular updates of its status along with updates of the database are to be provided through the lab's website (hmco.enpc.fr). The

web page https://hmco.enpc.fr/portfolio-archive/rw-turb/ already contains links to the summary calendars for past and ongoing measurement campaigns (daily updates). Database will continue to be organized as presented in this paper, and user interested by longer series once available can contact the corresponding author.

*Competing interests.*    The authors declare that they have no conflict of interest.

*Acknowledgements.*    The authors greatly acknowledge financial support from the ANR JCJC RW-Turb project (ANR-19-CE05-0022-01).

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
