# Peer review of "Three months of combined high resolution rainfall and wind data collected on a wind farm 110 km South-East of Paris (France)"

_Earth System Science Data, 2021_

## Author Comment (AC1)

Authors would like to thank the reviewer for its careful reading, comments and inspiring suggestions. We took them into account in the revised manuscript. Hopefully, you will be satisfied with it. Please find below a point by point answer to your comments.

This paper describes a dataset of very high resolution wind and rainfall data collected at a wind farm location in central France. A thorough description of the measurement site and equipment is given, followed by details of the data collected and some analysis on the potential limitations of the instruments when measuring at high frequencies.

From what I can see the data is complete and well documented on the zenodo repository. The inclusion of the python scripts and quick looks are a very useful addition for those wishing to quickly explore the data. I would comment that having to download the whole 9.7GB of data at once is an issue that may inhibit the use of the data to those who do not need everything. You comment in the manuscript that the large raw data files would only be needed by an expert user. Future datasets could be potentially be grouped by beginner and expert data users to improve accessibility of the data, and potentially improve the uptake of it's use.

Following you comment, if you consider that it is a satisfactory situation, we suggest to update the repository so that user can download each folder separately according to their need.

At present although I believe the dataset to be complete and well documented the manuscript quality is currently not high enough for publication. This could be improved by implementing the following comments.

Main comments:

- It would be helpful to mention the location of the wind turbine in the title and abstract of the paper as knowledge of the climatic region the turbine is located in would be very useful for those wanting to use the dataset. The time period the data is collected for would also be useful in the abstract.

The title was updated to "Three months of combined high resolution rainfall and wind data collected on a wind farm 110 km South-East of Paris (France)". Abstract was updated to account for your comments.

- Throughout the text there are numerous spelling and grammar errors which made it quite difficult to interpret the key messages. I would suggest a thorough proof read of the document to pick up on these. A number are highlighted in the minor comments below.

This was checked. Thank you for your careful reading.

- Ending the manuscript with a summary section of the data and details of some potential use cases would be helpful. Are there other applications as well as for the wind industry where these observations could be useful? How is this dataset better than others mentioned in the discussion that are already available?
- From what you say in Line 22, can you demonstrate in this paper how the rain rate impacts the conversion to wind power in this paper? This would further highlight the usefulness of

these measurements. Adding a section demonstrating this would significantly increase the value of the paper.

The paper is a data paper that aims at presenting in details a data set made available to the community. It does not aim at fully exploiting the data set for scientific studies, which will be done in further dedicated papers by the authors or community members using this data set. The data set was collected in the framework of an application to wind energy, but (as you suggest) potential applications of such high resolution rainfall and wind measurement campaign are much wider. There are notably applications in the field of hydrology.  Following your comment, a full paragraph was added in the introduction to clarify the purpose of the paper to avoid any misunderstanding. It also includes comments on potential applications in the field of hydrology.

- Within section 2.5 can you put the measurement period into more context. In terms of wind energy generation is there a large seasonal cycle at this location? And do you know what point in the cycle this is, and whether it is a particularly high/low wind year based on the large scale circulation conditions? Can you also comment on why there are large differences between the disdrometers and the stations?

Using 30 years of 50 m wind MERRA (Modern-Era Retrospective Analysis for Research and Applications) data, which is a NASA reanalysis (Bosilovich et al., 2018; Gelaro et al., 2017), the average daily wind was computed. Results are displayed in blue in the figure below. Higher winds are usually observed during winter (with daily average of 7.5 m/s) and lower ones during summer (with daily average of 4.5 m/s). The daily wind computed from anemometer #1 is in orange in the figure below (from 1/3 to 1/6) and correspond to a spring period. The available period has an average wind of 6 m/s, which is consistent with usual values; although such average fully neglects the variability which is what this data set enables to study.

Following your comment, Fig. 6 was updated to display temporal evolution of horizontal wind during the studied period. Previous comments with regards to common values in the area were also added to the manuscript.

[Figure]

The difference between rainfall estimates from disdrometer and stations is likely to be due to the fact that both devices rely on completely different measurement techniques. It should definitely be explored further in future investigations and this is now mentioned while commenting figure 6.

- I would remove the database structure from Section 3. Similarly for the bullet points in line 180-196 and 230-260. Alternatively this could all be moved to a supplement or changed to tables of variables available.

It could indeed be an option. However, authors believe that this would be easier for the user to have all this within the manuscript. Obviously, if you prefer, this can easily be changed. Bullet points have been changed to refer to panels of the updated figure 8 (see answer to a comment by the other reviewer).

Minor comments.
- Consider editing the spelling and/or grammar in lines 11, 13, 21, 31-33, 45, 75, 95, 115, 119, 124, 128, 207, 288, 299-301, 330.

This was done.
- Lines 15-19 Where are the previous studies that have looked at rain rates around wind turbines based? The climatic region that the wind turbines are in will be important for this relationship and would be worth commenting on (for example whether all in tropical or extra-tropical regions).

Outside experiment by Corrigan et al. was carried out in Ohio, USA; and this is now clarified in the manuscript.

- Line 29: Can you comment on the complexity of the atmospheric boundary layer and how that will impact the wind turbines?

Authors were simply referring to the fact that this is an area of increased complexity due to the interactions with the ground. This was added to the manuscript.

- Line 42: Can you give the dates of the measurement period?

This was added.

- Line 52: Figure 3 is mentioned before Figure 2.

Order was reversed.

- Figure 2: are there photo credits required here for the publication of the images?

In it now Fig. 3. Pictures were taken by 1st author and this was added in the caption.

- Figure 3: Can you include what the different colours mean in the caption. You could also possibly include the prevailing wind direction for some context and comment in the text on if it is influenced by local orography.

The meaning of colours was added in the caption. Prevailing wind is actually already displayed through the wind rose in Fig. 6.

- Line 73, define U_L

It is "the wind velocity along the corresponding axis". It is actually already mentioned at the beginning of the paragraph.

- Line 88 'Built-in' rather than 'Build-in'.

This was corrected.

- Line 94: What does OTT stand for?

OTT is the name of the manufacturer and this was clarified.

- Line 103: Check the display of drop size distribution information. The mixture of italics and normal font is confusing.

Italics is used for formulas and normal font for units. Should it be changed ?

- Section 3: You oscillate between the use of database and data base, please check for consistency.

Database is now used everywhere.

- Section 3.6 can you give an indication on how long the python code takes to run?

On a standard laptop, it typically takes few seconds to extract and display all the data for one day. This was added at the end of section 3.6.

- Line 343: Three month, rather than two month field campaign.

This was corrected. Thank you for your careful reading!

---

## Author Comment (AC2)

Authors would like to thank the reviewer for its careful reading, comments and inspiring suggestions. We took them into account in the revised manuscript. Hopefully, you will be satisfied with it. Please find below a point by point answer to your comments.

In this paper, the authors present a three-month long atmospheric measurements dataset from a meteorological mast installed at a wind farm in France. The dataset comes out of a campaign using two 3D sonic anemometers, two meteorological stations, and two disdrometers.

While reviewing this data paper, I came across some issues, which need to be addressed before they can be published.

Major comments:

1. The authors include the python scripts along with the dataset which is really helpful for the potential users for their analysis. That said, I feel that a single dataset file is presented which is too large to download (9.6 GB) and view if one wants to check just a file of interest. For better accessibility, it would be better to perhaps break the dataset into multiple subsets, for example by each device, or with a better strategy for sharing and accessing the data with ease.
Following you comment, if you consider that it is a satisfactory situation, we suggest to update the repository so that user can download each folder separately according to their need.

2. A brief description of how the updates/curation of the dataset will be handled is important to include in the manuscript.
As mentioned the campaign is still ongoing. The data set will continue to be organized as presented to this paper. Users interested by longer series are invited to contact the corresponding author. Following your comment, this was clarified in the "data availability" section.

3. What is the value of such field campaigns and data measurements on wind and rainfall? The answer to this question is missing in the manuscript.
4. The premise of this dataset is built around exploring the impact of rainfall on wind energy as you mention right at the beginning of your manuscript. The manuscript, however, does not touch anywhere on this.
The paper is a data paper that aims at presenting in details a data set made available to the community. It does not aim at fully exploiting the data set for scientific studies which will be done in further dedicated papers by the authors or community members using the data set. The data set was collected in the framework of an application to wind energy, but potential applications of such high resolution rainfall and wind measurement campaign are much wider. There are notably applications in the field of hydrology.
Following your comment (as well as one of the other reviewer), a full paragraph was added in the introduction to clarify the purpose of the paper to avoid any misunderstanding. It also includes comments on potential applications in the field of hydrology.

5. Most of the figures can be generally improved in terms of their size, presentation, and texts included. More detailed suggestions are made wherever relevant in the minor comments.

We implemented your suggestions.

6. Will there be any difference in the multifractal analysis results if one uses the entire 3-month dataset as opposed to 1 month dataset only? My concern is primarily from the seasonality point of view.
The other months were tested and gave the same results with regards to the studied issue in this data paper. Same overall results were obtained from analysis with 1 day, 1 week or 1 month considering either rainy or dry periods. Further analysis of the retrieved data more focused on physical interpretation of the processes will be carried out in further investigations and presented in a paper not corresponding to a data paper. This was clarified in the manuscript.

We noticed that UM and spectral analyses displayed were done on data from Jan/Feb 2021 and not from the 3 months of data that's shared with the paper. This was because the scaling analyses were performed with data from initial stage of campaign, before finalizing the dataset to be shared alongside the manuscript. To be more consistent, we have updated Section 4 with one-month data from provided database (01/03/2020 to 01/04/2020). Results are similar to the ones reported before which also confirms the uniformity of database.

Minor comments:

7. The abstract should include the location where the data was taken. Also, including a sentence on the target users of this dataset would benefit the readers.
This was done.

8. L5-6: The sentence doesn't read well. Rephrase it!
This was corrected.

9. Check parenthesis in the Figure 2 caption.
This was corrected.

10. L61: Add a sentence or two further describing the terrain settings around the mast.
As you suggested, this was done.

11. L66: "of" missing in the sentence after "are located in one ……"?
This was corrected.

12. Add North arrows in Figures 3 and 4. In figure 4, instead of the elevation product in the legend, use "Elevation [m]".
North arrows were added to Figures 1, 3 and 4 as suggested. Elevation was indicated in the legend, and the product name is now only in the caption.

13. L75: Begin the sentence with a lowercase "which".
This was corrected.

14. L108: There are several pieces of literature on this topic estimating rain rate from disdrometer data that are worth noting here. Include some key ones.
Some references were added as you suggested.

15. L112: Are you referring "Anemometer #2"?
Yes indeed, thank you for spotting this.

16. Figure 6: I suggest rearranging this figure with the R vs time plot in the first row while the rest in the second row. This might provide a better distinction of rain rate between each station/disdrometer.
Following your comment, we tested this and given that three months of data are represented, it did not enable to improve visibility of the various curves. The main purpose of this graph is not to look at details for an event (quicklooks are here for this purpose) but to highlight the periods of rainfall and maximum observed peak. However, given that the "stations" usually provide lighter rainfall than the "disdrometers", we decided to plot them at the end which slightly improves visibility. The size of the figure was also slightly increase to fit the whole page. Temporal evolution of the wind was also added following comments by the other reviewer.

17. L176: Remove "the" after "This is done through ….".
This was corrected.

18. L179: Correct the sentence as "It provides a summary….".
This was corrected.

19. Figure 8: It needs many details to be easily comprehensible to the readers. Each of the subplots should be labeled and properly described in the caption or referred to in the description in the text. Also, you have enough space to make it bigger for readability.
Panels are now numbered, and the corresponding referencing has been updated in the text to improve readability. We agree with you with regards to the size, and it will be handled by the journal team on the final version.

20. L226: Check the sentence for correctness.
This was corrected.

21. L271: Should it be "more than…"?
Indeed you are correct and it was changed.

22. Figure 9: This figure should follow the text describing it rather than the other way around. Also, in the caption, there is no subplot (d) in the figure. Make sure you add it.
Indeed it was missing. It has been added.

23. L313: The paragraph should include a brief description of what Trace Moment (TM) analysis is and how it can support the Universal Multifractal analysis.
As you suggested this was added at the beginning of the paragraph.

24. L319: Maybe you could use a column here to remove confusion on whether it is a minus sign.

Indeed, thanks for the suggestion.

25. L339: You mean "…It is….?
This was corrected.